# Joint spatiotemporal modelling reveals seasonally dynamic patterns of Japanese encephalitis vector abundance across India

**Lydia H. V. Franklinos**[1,2]*, **David W. Redding**[3], **Tim C. D. Lucas**[4], **Rory Gibb**[5,6], **Ibrahim Abubakar**[2], **Kate E. Jones**[1,3]

**1** Centre for Biodiversity and Environment Research, University College London, London, United Kingdom, **2** Institute for Global Health, University College London, London, United Kingdom, **3** Institute of Zoology, Zoological Society of London, London, United Kingdom, **4** School of Public Health, Imperial College London, London, United Kingdom, **5** Centre for Mathematical Modelling of Infectious Diseases, London School of Hygiene and Tropical Medicine, London, United Kingdom, **6** Centre on Climate Change and Planetary Health, London School of Hygiene and Tropical Medicine, London, United Kingdom

\* lydia.franklinos.16@ucl.ac.uk

**Data Availability Statement:** The vector data underlying the results presented in the study are

## Abstract

Predicting vector abundance and seasonality, key components of mosquito-borne disease (MBD) hazard, is essential to determine hotspots of MBD risk and target interventions effectively. Japanese encephalitis (JE), an important MBD, is a leading cause of viral encephalopathy in Asia with 100,000 cases estimated annually, but data on the principal vector *Culex tritaeniorhynchus* is lacking. We developed a Bayesian joint-likelihood model that combined information from available vector occurrence and abundance data to predict seasonal vector abundance for *C. tritaeniorhynchus* (a constituent of JE hazard) across India, as well as examining the environmental drivers of these patterns. Using data collated from 57 locations from 24 studies, we find distinct seasonal and spatial patterns of JE vector abundance influenced by climatic and land use factors. Lagged precipitation, temperature and land use intensity metrics for rice crop cultivation were the main drivers of vector abundance, independent of seasonal, or spatial variation. The inclusion of environmental factors and a seasonal term improved model prediction accuracy (mean absolute error [MAE] for random cross validation = 0.48) compared to a baseline model representative of static hazard predictions (MAE = 0.95), signalling the importance of seasonal environmental conditions in predicting JE vector abundance. Vector abundance varied widely across India with high abundance predicted in northern, north-eastern, eastern, and southern regions, although this ranged from seasonal (e.g., Uttar Pradesh, West Bengal) to perennial (e.g., Assam, Tamil Nadu). One-month lagged predicted vector abundance was a significant predictor of JE outbreaks (odds ratio 2.45, 95% confidence interval: 1.52–4.08), highlighting the possible development of vector abundance as a proxy for JE hazard. We demonstrate a novel approach that leverages information from sparse vector surveillance data to predict seasonal vector abundance–a key component of JE hazard–over large spatial scales, providing decision-makers with better guidance for targeting vector surveillance and control efforts.

available from: https://figshare.com/s/377b76b6b79ffa2561cf. This dataset includes all vector data collected including records that pooled observations for more than one month. Sources for all freely available environmental datasets are described in S2 Table. Health data are available from the Ministry of Health & Family Welfare, Government of India: https://www.idsp.nic.in/index4.php?lang=1&level=0&linkid=406&lid=3689.

**Funding:** This research was supported by a Natural Environment Research Council (NERC) PhD studentship (https://london-nerc-dtp.org/) for LHVF (Grant ID: NE/L002485/1), an MRC UKRI/Rutherford Fellowship (https://stfc.ukri.org/funding/fellowships/ernest-rutherford-fellowship/) (Grant ID: MR/R02491X/2) and Wellcome Sir Henry Dale Fellowship (https://wellcome.org/) (Grant ID: 220179/Z/20/Z) (both DWR). IA acknowledges funding from the UK NIHR (https://www.nihr.ac.uk/) (Grant ID: NF-SI-0616–10037), EDCTP PANDORA Consortium (http://www.edctp.org/) and the Medical Research Council (MRC) (https://mrc.ukri.org/). KEJ acknowledges the Dynamic Drivers of Disease in Africa Consortium, NERC project no. NE-J001570-1 which was funded with support from the Ecosystem Services for Poverty Alleviation Programme (ESPA). The ESPA programme was funded by DFID (https://www.gov.uk/government/organisations/department-for-international-development), the Economic and Social Research Council (ESRC) (https://esrc.ukri.org/) and NERC (https://nerc.ukri.org/). RG was supported by a Graduate Research Scholarship from University College London. TL was funded by an MRC Centre for Environment and Health Fellowship (https://environment-health.ac.uk/) (Grant ID: MR/T502613/1). The funders had no role in study design, data collection and analysis, decision to publish, or preparation of the manuscript.

**Competing interests:** The authors have declared that no competing interests exist.

## Author summary

Japanese encephalitis (JE) is the leading cause of viral encephalopathy in Asia with an estimated 100,000 annual cases and 25,000 deaths. However, insufficient data on the predominant mosquito vector *Culex tritaeniorhynchus*–a key component of JE hazard–precludes hazard estimation required to target public health interventions. Previous studies have provided limited estimates of JE hazard, often predicting geographic distributions of potential vector occurrence without accounting for vector abundance, seasonality, or uncertainty in predictions. This study details a novel approach to predict spatiotemporal patterns in JE vector abundance using a joint-likelihood modelling technique that leverages information from sparse vector surveillance data. We showed that patterns in JE vector abundance were driven by seasonality and environmental factors and so demonstrated the limitations of previously available static vector distribution maps in estimating the vector population component of JE hazard. One-month lagged vector abundance predictions showed a positive relationship with JE outbreaks, signalling the potential use of vector abundance as a proxy for JE hazard. While vector surveillance data are limited, joint-likelihood models offer a useful approach to inform vector abundance predictions. This study provides decision-makers with a more complete picture of the distribution of JE vector abundance and can be used to target vector surveillance and control efforts and enhance the allocation of resources.

## Background

Mosquito-borne diseases (MBDs) pose a substantial global health concern due to their ongoing geographic expansion and increasing incidence [1, 2]. Identifying hotspots of MBD risk is critical in informing effective interventions and safeguarding public health [3]. This is particularly important for understudied diseases, such as neglected tropical diseases, because resources for disease surveillance and control are often limited [4]. Mosquito-borne disease risk can be understood as the likelihood of an outbreak due to exposure of a susceptible population to an infected mosquito vector (hazard) [5]. Defining areas of MBD hazard requires knowledge of pathogen prevalence in reservoir host and vector populations however, these data are often not available. Therefore, models that predict how vector populations may vary over space and time, thereby estimating a key component of hazard, have become vital tools in MBD epidemiology [6, 7]. Nevertheless, considerable costs associated with vector sampling [8] have resulted in the limited availability of long-term vector surveillance datasets over large spatial scales, hindering the ability to predict vector abundance accurately and inform interventions.

Vector abundance i.e., the number of individuals in a site at a given time, and seasonality i.e., intra-annual change in abundance, are important contributors to pathogen establishment, persistence and transmission [6, 8, 9]. For example, regions with high vector abundance and a low seasonality (i.e., longer periods when adult vectors are active) will lead to increased likelihood of pathogen establishment and persistence [8]. Longer periods of high vector abundance may also increase the likelihood of pathogen transmission between vectors and hosts due to increased contact rates that could lead to pathogen exposure i.e., via vector feeding [8, 10]. Despite the epidemiological importance of vector abundance, most commonly available vector surveillance data consist of categorical information on occurrence (i.e., presence/absence) and rarely provide quantitative information on abundance [11].

The relative availability of vector occurrence data has contributed to the popularity of species distribution models (SDMs) in MBD research [6, 7, 12]. These statistical models typically correlate the presence of a species at multiple locations with environmental covariates to predict species distributions [13]. Although they provide valuable information on potential vector geographic distributions, knowledge of where vectors can occur is insufficient to provide an accurate estimation of MBD hazard [8] particularly because these models do not consider spatial and temporal dynamics [14]. In addition, for widely-used SDM approaches such as boosted regression tree (BRT) models and MaxEnt, uncertainty estimates are produced by bootstrapping data which can be computationally prohibitive [15, 16]. Without predictive uncertainty metrics, results may be misleading for decision-makers since it may be difficult to distinguish between regions with accurate predictions and those that have a high degree of uncertainty [17]. Alternatively, seasonal vector abundance has been estimated using mechanistic models of vector populations based on a system of differential equations depicting each life stage [10, 18]. However, these models rely on large amounts of experimental or empirical data [6] which can be expensive to obtain and are often sparse for many vector species [19]. The lack of long-term abundance data [1, 9] has also meant that statistical models of seasonal vector abundance often exist for local [20–22] rather than for national or regional geographic scales. Overall, there is a need for improved estimates of components of MBD hazard which also account for uncertainty to enable a better understanding of seasonal patterns in the risk of disease transmission.

One of the most important yet relatively understudied MBDs is Japanese encephalitis (JE), the leading cause of viral encephalopathy in Asia [23–25]. JE accounts for over 100,000 human cases and 25,000 deaths annually, primarily affecting children and those living in rural, agricultural areas [25, 26]. Although the disease is endemic in 24 countries [25], the majority (87%) of cases in Asia are reported from India, Nepal, China and Vietnam [27, 28]. The causative pathogen, Japanese encephalitis virus (JEV) is maintained in an enzootic transmission cycle between mosquitoes and a range of amplifying hosts including domestic pigs and ardeid wading birds [29]. Agricultural practices such as rice cultivation and pig breeding provide an ideal environment for human exposure to JEV, however other factors such as population immunity due to vaccination will also influence the risk of disease outbreaks [30]. The virus is predominantly transmitted by the mosquito vector *Culex tritaeniorhynchus* Giles, 1901 (Diptera: *Culicidae*) [31] and JE outbreaks are reported to be strongly associated with vector abundance [32–34]. Despite *C. tritaeniorhynchus* being a major threat to human health and wellbeing, there are limited surveillance data for this species [35] which has impeded knowledge on spatiotemporal trends in vector abundance, a constituent of JE hazard.

*C. tritaeniorhynchus* population dynamics are strongly linked to climatic conditions, such as temperature and rainfall [36, 37], and to anthropogenic activities that increase standing water, such as irrigated agriculture [38–41]. Experimental studies on other *Culex* species have found important life history traits such as development rate and survival generally peak at 15.7–38.0˚C (mean thermal optimum = 28.4˚C) and then decline to zero for thermal minima (mean = 9.5˚C) and maxima (mean = 39.5˚C) [19]. Rainfall can both positively influence *C. tritaeniorhynchus* abundance via the creation of standing water for vector breeding [37, 42, 43] and negatively impact abundance during the monsoon [44] via the destruction of breeding sites [45]. Irrigated agriculture provides suitable habitat for vector development and *C. tritaeniorhynchus* is reported to breed preferentially in rice paddy fields [38, 39]. Indeed, previous studies have shown that vector abundance is positively associated with rice field density [46], rice crop growth stage [40, 41] and standing water availability [38, 47]. Interestingly, the availability of standing water due to irrigation practices may lead to a reduction in vector seasonality (i.e., by extending vector breeding seasons), especially in arid regions which would

otherwise be unable to sustain vector development during summer months [40, 41, 48–50]. Although environmental conditions are known to underpin the seasonal dynamics of many vector populations [18, 51], the importance of these factors in driving broad-scale spatial and temporal patterns of JE vector populations remains poorly defined.

Previous studies have investigated the spatial distribution of *C. tritaeniorhynchus* occurrence using SDMs [35, 52–54] however, there is a paucity of data on seasonal vector abundance. Bayesian hierarchical modelling approaches have been used widely for other animal species to estimate biodiversity trends by integrating multiple data types in a single estimator [55, 56]. This joint-likelihood approach has also been used in MBD research to explicitly account for differences in data quality and structure (i.e., different probability distributions) and can handle and quantify sources of uncertainty associated with each data type [57, 58]. Here, we use this approach to develop a joint-likelihood Bayesian hierarchical model that leverages spatial information from vector occurrence probability to estimate seasonal vector abundance for principal JE vector, *C. tritaeniorhynchus* across India. Firstly, our study aims to quantify the importance of different environmental drivers of *C. tritaeniorhynchus* abundance–a key component of JE hazard. We hypothesise that a critical driver of vector abundance is standing water provided by rice crop irrigation practices and periods of heavy rainfall during the winter and monsoon seasons. Secondly, we aim to construct seasonal vector abundance maps for India that account for uncertainty in predictions. Thirdly, we use logistic regression to test whether there is a relationship between mosquito abundance estimates and JE cases and discuss the potential for vector abundance to be used as a proxy for JE hazard. The purpose of this research is to provide decision-makers with useful information that will assist in their resource allocation for intervention strategies and highlight areas to target for future vector surveillance. India is used as a case study since it has one of the highest JE burdens in Asia [26–28] and reports both endemic and epidemic epidemiological patterns [59, 60].

## Materials and methods

### Datasets

**Vector data.** We assembled a database of geo-referenced, spatially, and temporally unique *C. tritaeniorhynchus* vector occurrence and abundance records in India from published literature. A systematic literature search was conducted in PubMed and Web of Science using the search terms "*Culex tritaeniorhynchus*" and "India". The search was limited to articles published in English between 1st January 1990 and 31st December 2017 and returned 101 unique citations. Article abstracts were screened to meet the following criteria for inclusion; (i) the reported study was undertaken after 1990, (ii) surveys provided species-level information at the studied location, and (iii) the surveys were conducted in the mainland of India. The full text articles were then reviewed and excluded if they pooled observations for more than one month since this would increase uncertainty in the associations between vector occurrence and abundance and predictor variables. The resulting 24 studies that met the inclusion criteria were used to build the dataset. The database included 340 unique records of adult vectors which ranged from 1990–2012 from 57 sampling locations resulting in data from 352 location-months (see S1 Table). Of the 340 unique records, 74 were occurrence-only records and 266 included occurrence and abundance data (Fig 1). Records that included occurrence and abundance data were used twice in the analysis; once as occurrence data and once as abundance data (total occurrence data n = 340, total abundance data n = 267) (see S1 Table). The study period was chosen to maximise the number of vector surveillance records whilst enabling the use of high-resolution land cover datasets that were available from 1990s. We built on previous

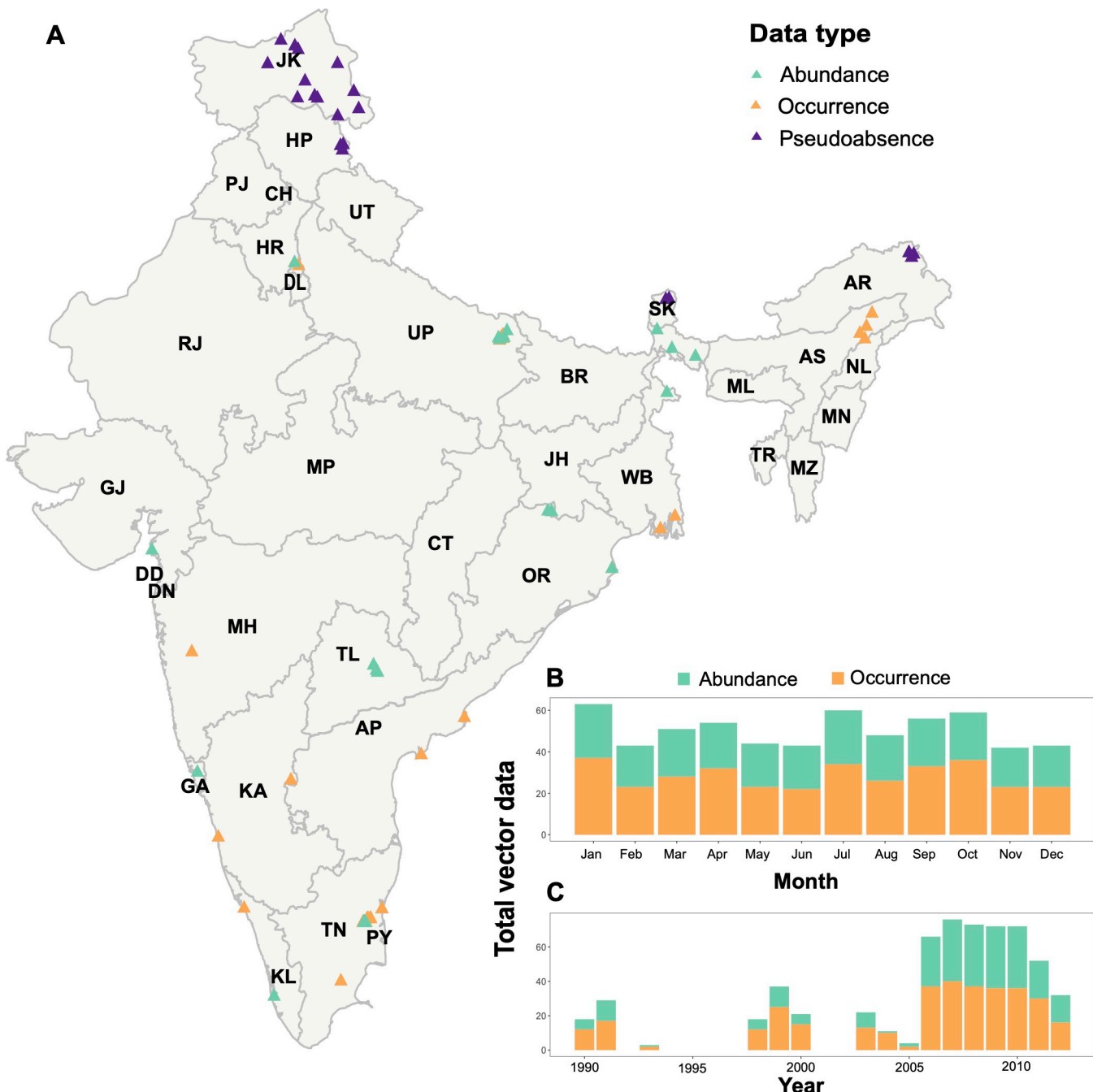

**Fig 1. Spatial and temporal distribution of vector surveillance dataset used in model.** (A) Points show the geographical sampling locations (n = 57) of the C. tritaeniorhynchus records across India*, with occurrence-only records coloured orange (n = 74), records which included occurrence and abundance data in green (n = 266), and pseudoabsence records in purple (n = 20). Stacked barplots show the temporal distribution of the total vector occurrence (orange) and abundance data (green) used in the analysis per month (B) and year (C). *Abbreviations for Indian states and union territories: AP—Andhra Pradesh, AR—Arunachal Pradesh, AS—Assam, BR—Bihar, CH–Chandigarh, CT- Chhattisgarh, DD—Daman and Diu, DL—Delhi, DN—Dadra and Nagar Haveli, GA–Goa, GJ–Gujarat, HP—Himachal Pradesh, HR—Haryana, JH—Jharkhand, JK—Jammu and Kashmir, KA—Karnataka, KL–Kerala, MH—Maharashtra, ML—Meghalaya, MN—Manipur, MP—Madhya Pradesh, MZ—Mizoram, NL—Nagaland, OR—Odisha, PJ—Punjab, PY—Puducherry, RJ—Rajasthan, SK—Sikkim, TL–Telangana, TN–Tamil Nadu, TR–Tripura, UP—Uttar Pradesh, UT—Uttarakhand, WB–West Bengal. Source of base layer https://gadm.org.

*C. tritaeniorhynchus* occurrence datasets developed by Miller *et al.*, [52] and Longbottom *et al.*, [35] to include information on mosquito presence, absence, and abundance, collection method, collection year and month, and habitat descriptions. Mosquito sampling locations in each study were identified as point locations. We calculated effort-corrected abundance values of *C. tritaeniorhynchus* from the raw measurement values by aggregating monthly counts and standardising them to survey effort (one survey hour) abundance measure for each month. Most abundance data (86%; n = 228) were recorded from the state of Tamil Nadu (Fig 1A) and only four studies performed continuous abundance measurements over consecutive months (see S1 Table). Survey effort (one survey hour) vector abundance measures were transformed to logscale to conform to normality and ranged from 0 to 6.49 (0 to 655 true scale) with a mean of 3.61. The occurrence and abundance data used in the models were evenly distributed across all study months (Fig 1B). However, there is a lack of vector data from 1992 to 1998 and most abundance data were recorded from 2006 to 2012 (Fig 1C).

**Additional inferred absence vector data.** We randomly generated additional absence data for regions above 3500m since to our knowledge, this is above the altitude that *C. tritaeniorhynchus* mosquitoes have been recorded [61]. To limit artefactual spatial and temporal autocorrelation in model residuals, we limited these data to a total of 20 records from 12 locations which were randomly selected from high altitude regions in the states of Arunachal Pradesh, Himachal Pradesh, Jammu and Kashmir and Sikkim (Fig 1A) and randomly assigned a date from the study period.

**Seasonal, environmental and land use data.** We selected environmental variables hypothesised or reported to influence the presence or abundance of *C. tritaeniorhynchus* populations (see S2 Table and S1 Fig). For instance, temperature is known to influence the development and survival rates of mosquito vectors and the availability of standing water provided from precipitation or irrigated agricultural practices is required for mosquito breeding [41, 50, 62]. The full suite of covariates tested across all analyses, data sources and associated hypotheses, including those considered but then dropped from the model, are described as follows:

Climate variability was incorporated through inclusion of TerraClimate [63] high-spatial resolution rasters (1/24˚, ~4-km) for monthly cumulative precipitation (mm), monthly maximum and minimum temperatures ($^0$C). We calculated monthly mean temperature ($^0$C) from the maximum and minimum temperature datasets. Mean monthly precipitation was log transformed to represent the nonlinear effect reported between rainfall and vector abundance [64]. To represent the lag association between weather conditions and mosquito abundance [30], we also calculated average temperature and precipitation data for the two months prior to the vector observation (henceforth referred to as two-month lagged variables in this study) to account for the period for mosquito larval habitat to increase and the development period of the mosquito.

We obtained annual land cover data from the European Space Agency (ESA) Climate Change Initiative Land Cover dataset (version 3.14) for 1992–2012 (ESA; http://maps.elie.ucl. ac.be/CCI/viewer/index.php) with a spatial resolution of 300m. The 37 original land cover classes were reclassified into six broad groups (agricultural, mixed agricultural, forest, mixed vegetation, urban and water) since the land cover types associated with the vector surveillance data were not varied enough to evaluate the importance of more diverse land classes (i.e., rainfed versus irrigated cropland). Zonal statistics function was used to determine the percent cover of each of land cover class within 1km buffer around each location, with the buffer size based on previous analyses [65]. Since ESA land cover data were missing for 1990 and 1991, we assessed changes in the proportion of land cover classes for the period 1992 to 1995 and found strong significant correlation between the years (Mantel statistic R: 0.99, *p* = 0.001), so we used land cover data for 1992 for the missing years. Agricultural land use intensity can be

assessed via three categories: input metrics (e.g., irrigation), output metrics (e.g., yields) and system level metrics (e.g., actual vs. attainable yield) [66]. Due to the strong positive associations reported between *C. tritaeniorhynchus* abundance and rice paddy cultivation, we used the RiceAtlas database of global rice production [67] to extract district-level data for the agricultural intensification input metric of total annual rice area cultivated (hectares) and for the output metrics of total annual rice produced (tonnes) and average number of crops harvested per year. To assess seasonal variation in rice cropping practices, district-level data on the rice planting and harvesting months were also extracted from the RiceAtlas dataset.

All raster data layers were manipulated and resampled to a $0.208^0$ (~23km) grid cell size using a World Geodetic System 84 projection using the 'raster' package in R [68]. We examined all covariates for collinearity and excluded covariates that were collinear with one or more others (Pearson correlation coefficient >0.8).

**Japanese encephalitis human case data.**   Monthly JE human cases recorded were retrieved from the Indian Government's Ministry of Health and Family Welfare [69]. Data were obtained for the period January 2009 to December 2015 and were converted to geographic point locations (n = 123) from their village level description using online gazetteers (e.g., Google Maps). The data comprised of the number of confirmed cases rather than suspected cases since clinical signs for JE may overlap with several other diseases [70]. Confirmed cases correspond to those confirmed by laboratory tests using JE-ELISA on serum or cerebrospinal fluid samples.

## Statistical analysis

Statistical modelling was conducted using Bayesian hierarchical regression using Integrated Nested Laplace Approximation (INLA). This framework enables the development of spatiotemporal models that address data sparsity and spatial bias whilst also being computationally tractable [71, 72].

**Model specification.**   We developed a joint-likelihood Bayesian spatiotemporal model of *C. tritaeniorhynchus* with separate likelihoods for occurrence and abundance data. The first model tier estimates vector occurrence probability with species presence/absence (0, 1) as response $y_{pa}$ using a Binomial distribution with a logit link function, such that $p_i$ denotes the expected probability of vector occurrence and $n_i$ is the observed survey sample size at observation $i$:

$$y_{pa} \sim \text{Binom}(p_{i,} n_i) \tag{1}$$

$p_i$ is modelled as a function of environmental covariates and spatial, seasonal, and random effects:

$$\text{logit}(p_i) = \alpha + \alpha_{pa} + \sum_{k=1}^{K} \beta_k X_{k,i} + t_i + \gamma_i + u_i + v_i + \delta_i \tag{2}$$

where $\alpha$ is the intercept; $\alpha_{pa}$ is an occurrence data specific intercept; $X$ is a matrix of the environmental covariates at each observation, with vector of linear coefficients $\beta$; $t_i$ is a nonlinear effect for mean monthly temperature smoothed using a second-order random walk to represent expected nonlinear relationships between temperature and vector occurrence and abundance [19]; seasonality was included as an effect of reporting month specified as a second-order random walk ($\gamma_i$); and spatial variation was included using state-level spatially-structured (conditional autoregressive; $v_i$) and unstructured i.i.d. ($u_i$) effects jointly specified as a Besag-York-Mollie (BYM) model [73]. Finally, $\delta_i$ is an independent, identically distributed (i.i.d.) random effect of source study to enable the model to account for between-study variation in sampling effort that might otherwise confound inferences.

The second tier in the joint-likelihood model estimated relative vector abundance as response variable $y_{\text{abun}}$ using a Gaussian distribution such that $\mu_i$ denotes the expected mean of vector abundance with standard deviation, $\sigma$:

$$y_{\text{abun}} \sim \text{Norm}(\mu_{i,}\sigma) \qquad (3)$$

The same shared covariates and spatial, seasonal, and random effects parameters were included as for the first tier model apart from the occurrence specific intercept:

$$\exp(\mu_i) = \alpha + \sum_{k=1}^{K}\beta_k X_{k,i} + t_i + \gamma_i + u_i + v_i + \delta_i \qquad (4)$$

Prior to being included in the model, all continuous predictor covariates were standardised (to mean = 0, SD = 1) and log vector abundance was rescaled from 0–1 (to preserve zero as a reference point) to help with assigning model priors [74]. Weakly informative prior probability distributions (priors) were assigned for the intercept, $\alpha \sim N(0,0.6)$ and fixed effects, $\beta \sim N(0,0.3)$ to constrain the position and scale of the outcome of interest ($y_{\text{abun}}$) to fall within a reasonable range. The intercept for occurrence data $\alpha_{pa}$ is a single, fixed parameter that was only added in the first tier of the model when modelling occurrence data. It acts as a varying intercept so that all occurrence data are modelled as a separate cluster to abundance data and therefore allows some flexibility in the joint modelling of both data types. Fixed effects priors were centred on 0 to allow for positive or negative relationships between environmental covariates and vector abundance. We assigned penalized complexity (PC) priors [75] to hyperparameters of the month, state-level and study-level effects. PC priors were used to penalise the complexity resulting from deviating from a simple base model. The PC priors are defined such that the probability that a given hyperparameter ($\rho$) exceeds an upper limit ($\rho_0$) is $\chi$ (i.e., $P(\rho > \rho_0) = \chi$). The PC priors in the model include:

$$\begin{aligned} \textit{Seasonal effects}: \qquad & \text{P}(\rho_i > 0.05) = 0.01 \\ \textit{Unstructured state–level effects}: \quad & \text{P}(u_i > 0.175) = 0.01 \\ \textit{Study–level random effects}: \qquad & \text{P}(\delta_i > 0.175) = 0.01 \end{aligned}$$

These values were chosen by comparing the variance of the effect variables and the resulting difference in log vector abundance observed. For example, an i.i.d. effect with a SD of 0.175 would typically (95% probability interval) yield intercepts between -0.34 and 0.34. Transforming these values through a log link gives abundances between 0.71 and 1.4 and therefore the effect allows a variation in abundance of about 100%. We based the values on assumptions from the data that log vector abundance may vary by up to 33% between one month and the previous two months (order-two random walk), whereas it may vary by 100% between studies. A conservative PC prior (mean 0.5, precision 0.667) was assigned to the structured state-level effect to account for the assumption that the unstructured effect accounts for more of the variability than the spatially structured effect.

**Model selection.** Collinearity was detected between temperature variables therefore only monthly mean temperature was used in the final model to capture long term associations with vector abundance (i.e., reduced effect of temperature extremes). We conducted model selection on model covariates (all fixed and spatial, seasonal and study-level random effects), evaluating their contribution to the model fit by removing each component in turn from the full model and examining the effect on the Bayesian pointwise diagnostic metric Watanabe-Akaike Information Criterion (WAIC) [76]. We tested 17 environmental variables (see S2 Table). We screened variables using a single pass whereby we removed each variable in turn from the model and assessed the change in WAIC. Covariates that did not improve model parsimony

by a threshold of at least 2 WAIC units were excluded. We used this screening procedure to remove variables which were not improving model parsimony rather than searching for a best subset of variables as is performed in stepwise selection. The models were examined for fit and adherence to assumptions which included testing the model residuals for spatial autocorrelation using Moran's $I$ [77]. Temporal autocorrelation could not be assessed since the data were not sampled at regular intervals over the whole study period. In addition, to assess the influence of additional inferred absence data on model fit, we repeated the process of randomly selecting 20 inferred absence data points 25 times and examined the impact on WAIC.

We further evaluated the predictive ability of the models using random (10-fold) cross-validation which involved fitting separate models holding out data from each fold in turn. The random assignment of data to folds was chosen to represent the spatiotemporal variation in predictor space in all folds. The spatial clustering in abundance data meant that spatially structured cross-validation by state was not used for model evaluation [78]. The final model was selected by comparing models of increasing complexity, in terms of input variables and model structure, to a baseline model which only included spatial effects and study-level random effects. This baseline model represents static vector abundance predictions that do not account for seasonality. We compared the baseline model to a seasonal model which also included the addition of a seasonal effect to account for seasonality in vector abundance and an environmental model which included spatial, seasonal, and random effects and environmental covariates. The ability of the models to predict log vector abundance (unscaled) was compared using the mean absolute error (MAE) between the predicted posterior mean values and the corresponding observed log vector abundance [79] where lower values indicate a smaller difference between the predictions and the observations. In addition, we used conditional predictive ordinates (CPO) [80] and predictive integral transform (PIT) [81] as cross-validatory criterion for model assessment. For CPO, a value is computed for each observation with small values indicating a bad fitting of the model to that observation and the potential for it to be an outlier. Predictive integral transform provides a version of CPO that is calibrated so that values like between 0 and 1. A histogram of PIT values that appears approximately uniform indicates the model represents the observation well. We also compared the direction and magnitude of fixed effects for hold-out models to examine the robustness of vector-environment relationships. The fixed effects parameter estimates were assessed using the posterior mean and 95% credible interval which is interpreted as the interval that covers the true parameter value with a probability of 95%, given the evidence provided by the observed data.

**Spatiotemporal predictions of JE vector abundance and uncertainty.** The best-fitting model was used to predict seasonal relative vector abundance (logscale) per $(0.208^0)$ grid cell across India for the three main seasons: winter (October to February), summer (March to May), monsoon (June to September). The seasons were chosen for their distinct climatic characteristics with heavy rainfall in central regions and the eastern coast during the winter, heavy rainfall in southwestern and north-eastern India during monsoon and high temperatures with little to no rainfall during summer [82]. We evaluated the uncertainty in model predictions by mapping the SD in estimated vector abundance per grid cell for each season. A narrow SD (SD < 1) indicated low uncertainty and a wide SD (SD > 1) indicated high uncertainty.

**Model-outbreak data comparison.** To examine whether predicted mosquito abundance is correlated to JE cases, we compared observed human outbreaks of JE with model predictions for vector abundance at the same geographic location and calendar month. We define a JE outbreak as one or more confirmed or suspects cases of JE occurring in the same village within the same month. We converted JE outbreak data to binomial (presence/absence) data that a JE outbreak occurred in a particular geographic location and calendar month. We randomly generated pseudoabsence JE case data for 1000 locations for the 12 months (n = 12000) to assess

the ability of the model to correctly predict the probability that an outbreak occurred (which we describe as JE outbreak probability). We fitted a logistic regression of the probability of JE outbreak occurrence as a function of model-predicted vector abundance with and without a one-month lag using glm in R [83]. A null model (i.e., intercept only) was developed to represent predictions expected at random so that the effect of vector abundance predictions in explaining JE outbreaks could be assessed via comparing model Akaike Information Criterion (AIC) values. All data processing was conducted in R v.4.0.3 [83] with the packages R-INLA (http://r.inla.org) [84] and raster [68].

## Results

### Model selection

Table 1 shows model predictive accuracy statistics for a series of models of increasing complexity. The most complex model structure (Model 3), which contained spatial, seasonal, and random effects and environmental factors, achieved superior model fit (ΔWAIC from baseline model = -77.53) (and see S2 Fig). Comparison of out-of-sample predictive ability showed that the inclusion of seasonality in the model (Model 2) improved predictions of vector abundance by decreasing MAE by 15% (ΔMAE = -0.14) when compared to the baseline model (Model 1). The addition of environmental covariates (Model 3) led to a further 40% decrease in MAE when compared to seasonal Model 2 (ΔMAE = -0.33). As well as spatial, seasonal, and random effects, the final selected environmental model (Model 3) included six covariates after accounting for collinearity and covariate selection as described. The fixed effects in the final model included two-month lagged precipitation, proportion of land under agricultural use in 1km radius, annual number of rice crops, rice area cultivated, and rice produced per district and a nonlinear function for mean temperature. The CPO and PIT histograms demonstrated that addition of environmental covariates in Model 3 led to a better fit of the model to the data and a superior representation of the observations when compared to the other models (S3 Fig). Model residuals displayed no significant (p <0.05) spatial autocorrelation among sites. The random selection of inferred absence data points was found to have no substantial impact on the ΔWAIC values for the different models (S3 Table).

### Associations between environmental variables and vector abundance

We found that *C. tritaeniorhynchus* abundance was influenced by climatic and land use factors (Fig 2B). We found positive associations between vector abundance and two-month lagged precipitation, number of rice crops and annual rice production. The annual area under rice cultivation had a negative effect on vector abundance and the proportion of land under

**Table 1. Model selection results for models of increasing complexity.** The table details the structure of the joint-likelihood models and their corresponding within-sample predictive accuracy assessed on Watanabe-Akaike Information Criterion (WAIC) values. Best models were selected based on minimising WAIC while adhering to model assumptions. Out-of-sample predictive accuracy was compared using mean absolute error (MAE) statistic for random cross validation. Fixed effects included two-month lagged precipitation, proportion of land under agricultural use in 1km radius and district-level measures for annual number of rice crops and total rice area cultivated and rice produced per year. Mean temperature was included as a second-order random walk function to represent the nonlinear relationship between temperature and vector population dynamics. Non-environmental effects considered were for month (M) and state-level spatial (ST) effects specified as a BYM model and study-level (S) random effects.

| Model | | Non-environmental effects | Environmental effects | WAIC | MAE |
|---|---|---|---|---|---|
| 1 | Baseline model | ST, S | - | 722.15 | 0.95 |
| 2 | Seasonal model | M, ST, S | - | 651.14 | 0.81 |
| 3 | Environmental model | M, ST, S | Precipitation, Agri. land proportion, Annual rice crops, Annual rice area, Annual rice production, Nonlinear temp. function | 644.62 | 0.48 |

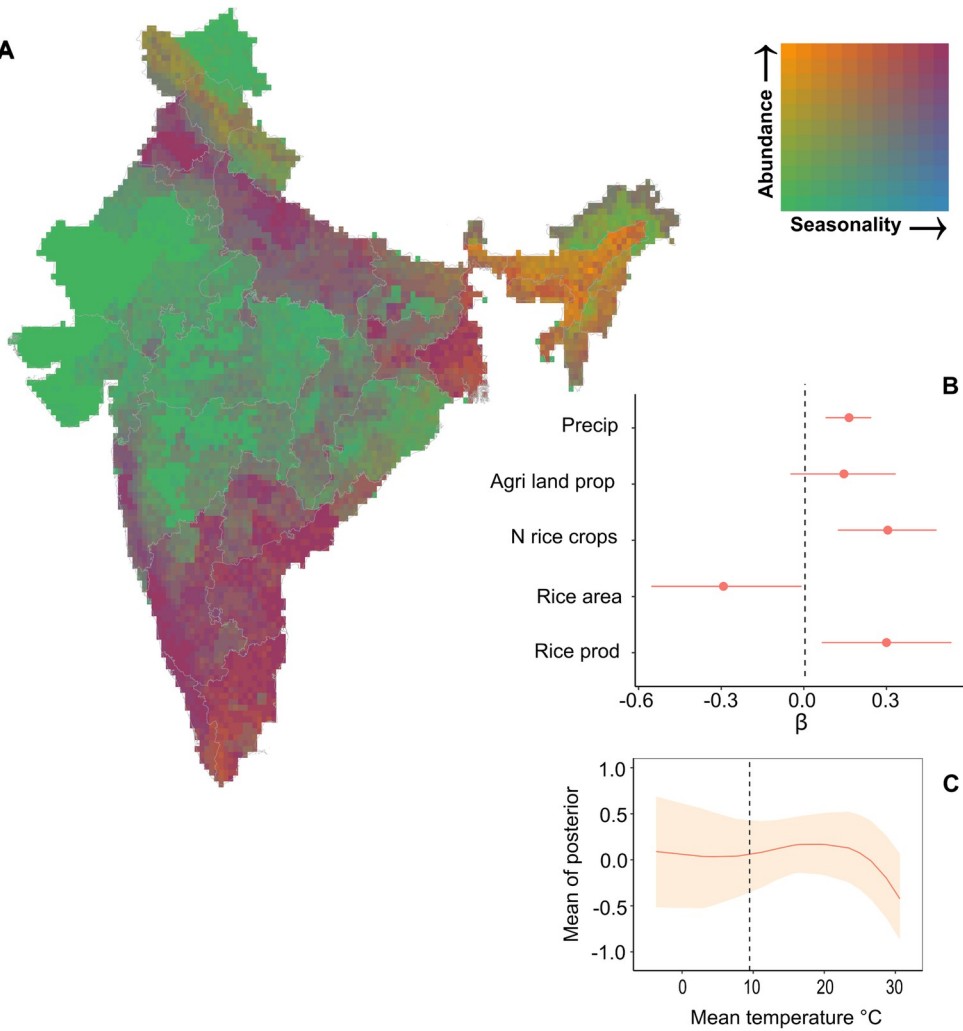

**Fig 2. Spatiotemporal correlates of JE vector abundance across India averaged over the period 1990–2012.** Map to show predicted *C. tritaeniorhynchus* abundance (maximum annual value) and vector seasonality (intra-annual variance in abundance) (A). These measures were calculated from the scaled abundance predictions and ranged from 0 to 7 logscale for maximum abundance and 0 to 3 logscale for seasonality. The map displays areas of high perennial vector abundance as orange, high seasonal vector abundance as pink, low perennial vector abundance as green and low seasonal vector abundance as blue. The fixed-effect parameter estimates and 95% credible intervals for the joint likelihood model (B) show that vector abundance is strongly influenced by climatic and land use variables. The nonlinear relationship between monthly mean temperature and vector abundance for the observed range of temperatures (C) where 95% CI is shown shaded and peaks at around 23°C and then declines. The reported thermal minima (9.5°C) for important *Culex* species life history traits [19] is indicated with a dashed line. Source of base layer https://gadm.org.

agricultural use had a weakly positive but uncertain association. Annual rice area and annual rice production had relatively wide credible intervals (CIs) for their parameter estimates when compared to the other covariates making the effect of these parameters on vector abundance more uncertain. These fixed-effects estimates were robust to randomly structured sensitivity tests (S4 Fig). We found that the inclusion of a nonlinear effect for mean monthly temperature without a lag improved model predictive ability when compared to the nonlinear effect with two-month lagged temperature (ΔWAIC = -81.83). The resulting temperature function suggests an increase in vector abundance from 9°C with a peak at around 23°C (Fig 2C). CI widths were low for this function at high temperature values.

### Spatiotemporal predictions of JE vector abundance and uncertainty

Spatially projecting the final model predictions revealed differences in predicted areas of high (i.e., hotspots) or low (i.e., coldspots) *C. tritaeniorhynchus* abundance between seasons (Fig 3). Peaks in vector abundance were found in the northern, eastern, north-eastern, and southern regions, with highest levels predicted during the winter months (October to February) and lowest levels during the summer months (March to May). Hotspots of vector abundance were predicted with low uncertainty (i.e., narrow SD) in northern, southern, and north-eastern India during the winter (Fig 3A) and in north-eastern and southern India during the summer (Fig 3B) and monsoon (June to September) seasons (Fig 3C). By contrast, hotspots were predicted with high uncertainty (i.e., wide SD) for all seasons in the northern state of Punjab, the eastern state of West Bengal and the south-eastern state of Andhra Pradesh. Areas predicted with low vector abundance (i.e., coldspots) were predicted throughout the year in the

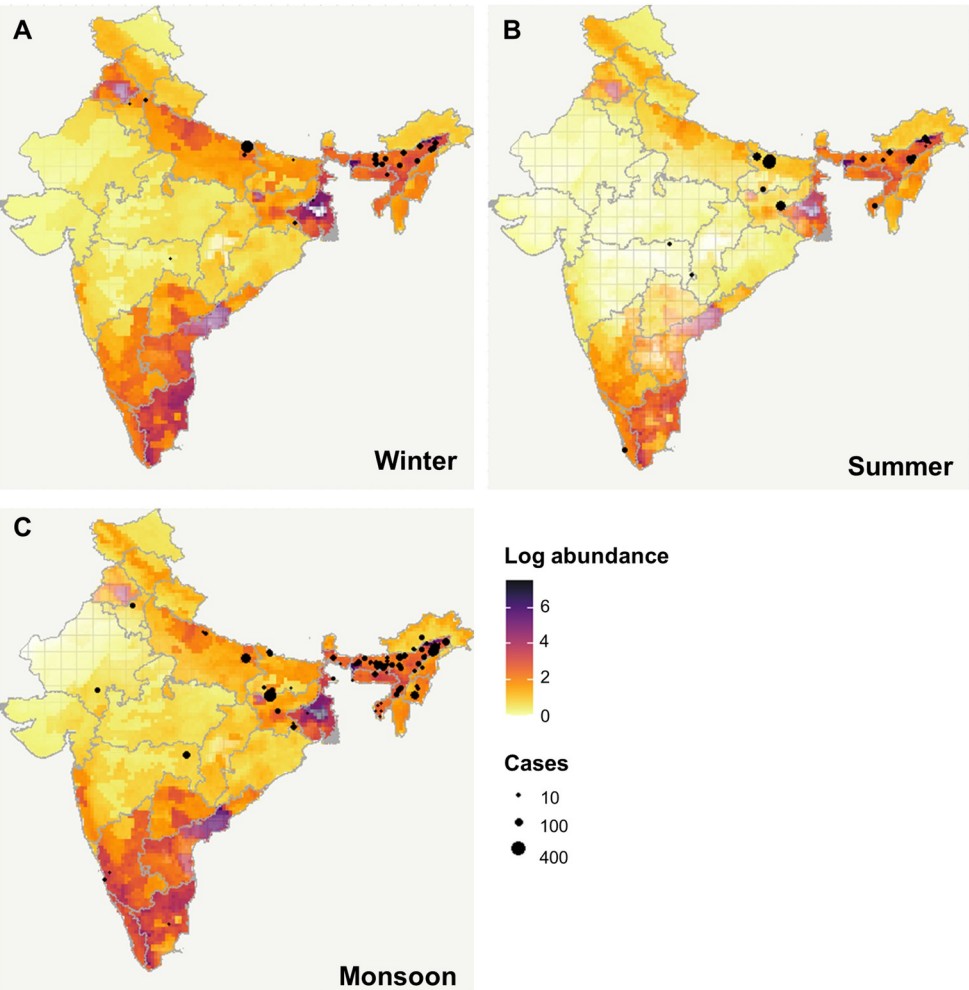

**Fig 3. Predicted seasonal abundance of *C. tritaeniorhynchus* across India for the period 1990–2012.** Average vector abundance (logscale) for the (A) winter (October to February), (B) summer (March to May) and (C) monsoon (June to September) seasons. The figure legend is scaled from 0 to 7 logscale, with light yellow colours signifying low vector abundance and dark purple emphasising high abundance. Uncertainty in predictions was estimated from standard deviation (range 0–2 SD) and is represented in the maps by transparency, (high uncertainty is more transparent). The black circles represent the location and magnitude (i.e., number of cases) for JE human outbreaks per season during the period 2009–2015 across India [68]. Source of base layer https://gadm.org.

Himalayas, and in central and north-western states, and eastern state of Odisha. Uncertainty in coldspot predictions was low for the Himalayas throughout the year (likely as a result of inferred absence data) whereas summer predictions for Odisha, central and north-western states and monsoon predictions for Rajasthan were more uncertain (represented as increased transparency in Fig 3). Assessing vector abundance and seasonality simultaneously reveals hotspots of high perennial vector abundance in north-eastern areas and the southern tip of the country (Fig 2A). Conversely, high seasonal vector abundance is predicted in northern and southern regions (Fig 2A).

## Model-outbreak data comparison

The results for the comparison between predicted mosquito abundance and JE cases is summarised in S4 Table. Logistic regression of JE outbreak probability as a function of model predicted vector abundance with a one-month lag month showed superior predictive ability (AIC = 144.17) when compared to the same analysis with vector abundance predicted in the same month as the outbreak (AIC = 147.66) and to the null model (AIC = 168.02). Both model-predicted vector abundance with and without a one-month lag had a significant positive effect on human JE outbreaks however, the lagged variable had a stronger association (odds ratio [OR] 2.45, 95% confidence interval: 1.52–4.08) than the variable without a lag (OR 2.25, 95% confidence interval: 1.35–3.74) (see S4 Table). Plotting predicted JE outbreak probability against log-scaled vector abundance for the best-fitting model (S5 Fig) illustrates that the strong association between these variables is non-linear and plateaus at high levels of vector abundance lagged by one month.

## Discussion

This study details a novel approach to predict spatiotemporal patterns in *C. tritaeniorhynchus* abundance–a key component of JE hazard—using a joint-likelihood modelling technique that leverages information from sparse vector surveillance data. We show that the addition of environmental covariates in the model substantially improved out-of-sample predictive ability, highlighting the importance of environmental and climate data in driving JE vector abundance. This provides strong justification for producing spatiotemporal vector predictions to focus future work efforts and build towards forecasts of JE risk. This framework provides a powerful and flexible method to define seasonal JE vector abundance over large spatial scales and assist in guiding future surveillance efforts where long-term and large spatial scale data are not available or could not be practically acquired. This analysis builds on previous correlative studies of *C. tritaeniorhynchus* which have mapped vector occurrence but have overlooked seasonal variation in population dynamics and have not accounted for uncertainty within the predictions [35, 52–54].

A distinct temporal pattern was observed across India in predicted vector abundance with peaks in the winter (October to February), reductions during the summer (March to May) and increased vector abundance again during the monsoon (June to September). This temporal pattern can be explained by seasonality in climatic factors during the year which supports findings in previous studies [37, 42, 85] and our hypothesis that vector abundance would be strongly influenced by seasonal rainfall. During the monsoon, heavy rainfall moving in a south-westerly direction across the country has been reported to enhance the availability of vector breeding habitats [44] and causes a reduction in local temperatures [86] which provide suitable environments for vector development. The peaks in vector abundance observed during the winter months probably reflect the post-monsoon rice cultivation period when water availability is high in the paddy fields [87]. This translates to the strong positive influence of

lagged precipitation on JE vector abundance found in this analysis and in other studies [30]. Conversely, high temperatures and low rainfall during the summer months probably limits vector survival and breeding [37], especially in areas with low levels of irrigated agriculture. Climatic conditions will also influence areas with predicted low perennial vector abundance such as arid regions in the northwest and northern states in the Himalayas which record temperatures beyond the thermal limit for *Culex* species vectors [19].

In addition to precipitation and temperature, land use and rice cultivation metrics were identified as important drivers of broad-scale spatiotemporal patterns of vector abundance. The importance of land use factors is illustrated by comparing hotspots of JE vector abundance in southern and north-eastern India which have high levels of irrigated agriculture despite differing climates (i.e., tropical in south, temperate in northeast) [88]. Regions with high proportions of agricultural land allocated to intensive irrigated agriculture provide suitable vector breeding habitats for extended periods which undoubtedly influence vector abundance and seasonality. Indeed, regions that cultivate rice biannually report lower vector seasonality compared with those that have a single annual crop [89]. The positive relationship between land use intensity metrics for rice crop cultivation (i.e., number of rice crops cultivated and amount of rice produced per year) and vector abundance detected in this study, supports previous research that has found a strong positive association between vector abundance and rice irrigation practices at local scales [38, 41, 46, 90]. Surprisingly, we found that the annual area under rice cultivation was negatively associated with vector abundance, albeit with wide CIs. This result may be spurious due to data quality issues or could be explained by unmeasured underlying factors such as agrichemical use (i.e., fertilisers and pesticides) [91], methods of irrigation (i.e., surface, sprinkler or drip irrigation) or use of fallow periods between crops which may lead to changes in local ecology (e.g., biotic interactions such as competition and predation) [92]. Indeed, local changes in ecology due to rice crop phenology are also likely to influence the presence of JE hosts since wading bird use irrigated rice paddies as feeding habitat [93] and fallow fields may be used to graze livestock. Understanding these relationships would require improved understanding of rice crop phenology together with biodiversity monitoring in rice fields. Our findings highlight the impact of land use practices on JE vector abundance which may have implications for the predicted expansion of flooded areas for rice cultivation needed to improve food security [38, 94] and the ongoing intensification of rice production in India [95].

Spatiotemporal patterns in JE vector abundance varied widely across India with seasonal hotspots predicted in northern, eastern, and southern regions and perennial hotspots predicted in north-eastern regions and the southern tip of India (Fig 2). These results support the spatial pattern in endemic regions of India which report particularly high endemicity in the states of Uttar Pradesh in the north, Bihar and West Bengal in the east, Assam in the northeast, and Tamil Nadu in the south [96]. In addition, vector abundance predictions reflected the described seasonality in JE transmission with increased outbreaks reported during the monsoon and winter seasons (Fig 3). However, predicted seasonal hotspots in the southeast did not correspond to high cases, which could reflect factors not accounted for in the analysis such as unmeasured environmental factors affecting transmission, spatial biases in different datasets or differing vaccination and vector control measures. In addition, it may also reflect the importance of vertical transmission for this disease which is selected for when there is seasonality in vector abundance [97]. We found a positive correlation between one-month lagged vector abundance predictions and the occurrence of human JE outbreaks when using a simple correlative analysis (S4 Table). This analysis assumes that the location of the vector abundance will also be the location in which exposure occurred which may be inaccurate. Indeed, to fully gauge the strength of this association and assess the usefulness of vector abundance as potential

proxy for JE hazard would require a more complex model that accounts for temporal and spatial autocorrelation in model residuals and uncertainty in the model. The development of a reliable proxy for JE hazard would be invaluable since data on pathogen prevalence in both animal reservoir host populations and vector populations that is required to define areas of JE hazard remains scarce. The further translation of hazard to disease risk requires additional knowledge about the potential exposure and susceptibility of human populations. For example, data on human demography, socioeconomics and vaccination coverage will provide information on contact with pathogens (exposure) and likelihood of infection (susceptibility) [5]. Furthermore, potential lags between peak vector abundance and human cases that occur due to transmission dynamics or timeliness of reporting need to be considered [98]. Indeed, future studies could extend this analysis by including further information on hazard, exposure, and vulnerability of human populations as well as any potential time lags to determine spatiotemporal predictions of JE risk [12].

A significant limitation of this study was related to the spatial and temporal biases of available *C. tritaeniorhynchus* surveillance data which is likely connected to the high costs associated with vector sampling studies [8]. Although data paucity leads to less accurate predictions in data-poor regions, we accounted for this by presenting the level of uncertainty within predictions on the vector abundance maps. Furthermore, it should be acknowledged that model predictions will not provide accurate data at the local level, instead they reveal broad scale ecological patterns that can help to direct future research efforts. In addition, the generation of additional absence data assumes that vectors do not occur at altitudes above 3500m which may need to be reviewed overtime with future surveillance studies and the influence of climate change [99]. This study highlights the need for improved vector surveillance for JE, with the potential for future surveillance efforts to be targeted in those areas with high predicted vector abundance to validate our results with independent data and improve predictions in areas that have not been surveyed. In addition, we find that despite JE vector abundance predictions being relatively focal (Fig 2), the spatiotemporal distribution of vector sampling in the data are more evenly distributed across India (Fig 1), suggesting that spatial bias is not driving model predictions (Fig 3). A further limitation of this study was the coarse spatial resolution of rice cultivation data used in the model [67]. The data were provided at district-level which may have been too coarse to detect an accurate relationship between land use intensity metrics and vector abundance [98] and may have prevented the detection of a correlation between vector abundance and rice cropping calendar data [40]. Future studies could explore the use of vegetation datasets such as normalized difference vegetation index (NDVI) at high spatial and temporal resolution to provide more accurate information on rice cultivation metrics [100] and rice crop phenology [101] in India. Investigating the lagged effects of these land use factors on vector abundance [30] may also help to elucidate the unexpected negative association between area for rice crop cultivation and vector abundance.

Despite these limitations, this work provides a framework to monitor and predict the seasonal abundance of JE vectors which will be crucial for public health bodies in their objective "to strengthen surveillance, (and) vector control" [96]. Current management for JE varies regionally across India depending on socioeconomic factors and whether areas have historically recorded high cases [96]. With ongoing environmental change, we believe the Indian public health bodies cannot afford to continue to focus their vector surveillance efforts on currently endemic regions, and instead need to establish a broader scanning surveillance system which can assist in developing early warning signals for predicting and mitigating JE outbreaks nationally. The maps produced in this study could be especially useful to guide public health actors in targeting future vector surveillance in understudied regions predicted to have high vector abundance with varying uncertainty. These data could then be used to inform the

model and improve and update predictions. Our work may also be used to improve the effectiveness of vector control measures especially in areas predicted high seasonal vector abundance, so that instead of being employed during JE outbreaks as is current practice [96], measures could be employed prior to an outbreak when vector abundance is high.

## Conclusions

In this study we provided i.e. scale estimates of the variation in vector abundance across space and time by leveraging different types of data sources for *C. tritaeniorhynchus*, an understudied JE vector. We showed that distinct spatiotemporal patterns of JE vector abundance were driven by seasonality and environmental factors and so demonstrated the limitations of previously available static vector distribution maps estimating vector occurrence across large geographic ranges [35, 52, 54]. In addition, we showed that model predictions of vector abundance were positively correlated with JE outbreaks, highlighting the possible development of vector abundance as a proxy for JE hazard. We propose that the joint-likelihood model used in our research will be easily adaptable for other mosquito vectors and enable other vector abundance estimations to be made from limited vector surveillance data. Furthermore, this novel approach can be used to help guide future vector surveillance programmes by targeting data collection. Understanding the timing and drivers of patterns in vector abundance and seasonality offers important insights into how and when intervention measures should be applied to reduce JE risk and how disease risk may vary with future environmental changes.

## Supporting information

**S1 Fig. Maps of covariates used in models.** (A) average mean temperature per month ($^0$C) (example given for the year 2005); (B) average precipitation per month (mm) (example given for the year 2005); (C) number of rice crop rotations per year (average for period 2010–12); (D) total annual rice area cultivated per year in hectares (average for period 2010–12); (E) total rice produced per year in tonnes (average for period 2010–12); (F) land use classes (example given for the year 2005). Source of base layer https://gadm.org.
(TIF)

**S2 Fig. Diagnostic plots for joint likelihood models; scatterplot of predicted versus observed vector abundance (logscale) data.** Plots show observed data against model predicted values, and the red line shows the expectation if observed equals predicted for each model: (A) baseline (spatial effects and study- level random effects), (B) seasonal (spatial, seasonal, and random effects), (C) environmental (spatial, seasonal, and random effects and environmental covariates).
(TIF)

**S3 Fig. Histograms of CPO and PIT values for joint likelihood models.** Plots show CPO and PIT histograms, with the red line indicating the level of the of the different values if their distribution was uniform: (A) baseline (spatial effects and study- level random effects), (B) seasonal (spatial, seasonal and random effects), (C) environmental (spatial, seasonal and random effects and environmental covariates).
(TIF)

**S4 Fig. Random spatiotemporal cross-validation of the final model.** We tested the sensitivity of fixed effects estimates to random (10-fold) subsampling. Points and error bars show posterior marginal parameter distributions for each hold-out model (median and 95% quantile range), with colour denoting hold-out group. Directionality and magnitude of fixed-effects

estimates are robust to all tests.
(TIF)

**S5 Fig. Association between one-month lagged vector abundance and predicted JE out-break probability.** Vertical axis displays model predicted JE outbreak probability, and vertical axis gives predicted vector abundance on the log scale. Smooth line highlights the non-linear relationship of JE outbreak probability to predicted vector abundance with a one-month lag.
(TIF)

**S1 Table. Vector surveillance data used in analyses.** The table includes the study from which the data were extracted, the state or union territory in India in which the survey was con-ducted, the year of the survey, the type of data collected, the survey method, the total number of months that were surveyed, the number of sampling sites per study and the total number of datapoints (occurrence and abundance) generated from the study.
(DOCX)

**S2 Table. Data and rationale for covariates included in analyses.** The table includes the sources and rationale (hypothesises) for inclusion of covariates in spatiotemporal models of vector abundance.
(DOCX)

**S3 Table. Impact of additional inferred absence data on selection results for models of increasing complexity.** The table details the structure of the joint-likelihood models and the difference between their corresponding within-sample predictive accuracy assessed on Wata-nabe-Akaike Information Criterion (WAIC) values when additional absence data are excluded. The differences (Δ) in WAIC from the baseline for the environmental and seasonal models are still equivalently large when compared to the ΔWAIC values when the additional absence data are included.
(DOCX)

**S4 Table Model comparison results for observed JE outbreaks AIC, odds ratio and 95% confidence intervals reported from logistic regression of JE outbreak probability as a func-tion of model predicted vector abundance Vector abundance predictions were generated from the final model with and without a one-month lag A null model (ie, intercept only) was developed to assess the ability of vector abundance predictions in estimating JE out-breaks when compared to predictions expected at random.**
(DOCX)

## Acknowledgments

The authors are grateful to Ella Browning, Lauren Enright for their valuable advice and to Sue Daly for her comments on previous versions of the manuscript.

## Author Contributions

**Conceptualization:** Lydia H. V. Franklinos, David W. Redding, Ibrahim Abubakar, Kate E. Jones.

**Data curation:** Lydia H. V. Franklinos.

**Formal analysis:** Lydia H. V. Franklinos.

**Funding acquisition:** Lydia H. V. Franklinos.

**Investigation:** Lydia H. V. Franklinos.

**Methodology:** Lydia H. V. Franklinos, David W. Redding, Tim C. D. Lucas, Rory Gibb.

**Project administration:** Lydia H. V. Franklinos.

**Resources:** Lydia H. V. Franklinos.

**Software:** Lydia H. V. Franklinos.

**Supervision:** David W. Redding, Ibrahim Abubakar, Kate E. Jones.

**Validation:** Lydia H. V. Franklinos.

**Visualization:** Lydia H. V. Franklinos.

**Writing – original draft:** Lydia H. V. Franklinos.

**Writing – review & editing:** Lydia H. V. Franklinos, David W. Redding, Tim C. D. Lucas, Rory Gibb, Ibrahim Abubakar, Kate E. Jones.

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
