## [Decision Letter · Decision Letter 0]

2 Sep 2021

Dear Dr Franklinos,

Thank you very much for submitting your manuscript "Joint spatiotemporal vector modelling reveals seasonally dynamic hazard patterns of Japanese encephalitis across India" for consideration at PLOS Neglected Tropical Diseases. As with all papers reviewed by the journal, your manuscript was reviewed by members of the editorial board and by several independent reviewers. In light of the reviews (below this email), we would like to invite the resubmission of a significantly-revised version that takes into account the reviewers' comments. 

We cannot make any decision about publication until we have seen the revised manuscript and your response to the reviewers' comments. Your revised manuscript is also likely to be sent to reviewers for further evaluation.

Sincerely,

Andrew S. Azman

Deputy Editor

Andrew Azman

Deputy Editor

Reviewer's Responses to Questions

**Key Review Criteria Required for Acceptance?**

**Methods**

-Are the objectives of the study clearly articulated with a clear testable hypothesis stated?

-Is the study design appropriate to address the stated objectives?

-Is the population clearly described and appropriate for the hypothesis being tested?

-Is the sample size sufficient to ensure adequate power to address the hypothesis being tested?

-Were correct statistical analysis used to support conclusions?

-Are there concerns about ethical or regulatory requirements being met?

Reviewer #1: Please refer to attached document

Reviewer #2: The objectives are clearly articulated. 

Key concerns on statistical methods and whether they support the conclusions:

1. Correction for mosquito sampling effort and trap types in the input data

Line 202-205 “We calculated effort‐corrected abundance values of C. tritaeniorhynchus from the raw measurement values by aggregating monthly counts and standardising them to survey effort (one person hour) density measure for each month”

I very much appreciate the efforts to correct for sampling effort. However, further brief explanation is needed of the adult trap types and methods used. Was there any correction performed for different adult trap types (e.g. CDC versus mosquito magnets), since these are known to pull in vectors over different ranges and attract widely differing abundance levels of mosquitoes. Do you mean “one trap hour” rather than “one person hour” or are these all human landing catches? 

2. Scaling of the input data and units of the mean absolute error

Line 335-337. It seemed strange to rescale the vector abundance data since this may obscure important biological information about hazard and size of model errors. I thought only rescaling of covariates was required to assist with assigning priors and model convergence? This is compounded later on when it is unclear whether the units for the mean absolute error (which always reflect the data) are in units of log abundance or units of the 0-1 rescaled log abundance? This makes it difficult for the reader to judge the biological significant of the difference in mean absolute error between the model with environmental factors and seasonal effects. It would be beneficial to analyse the conditional predictive ordinates from the model and the logarithmic score based on these as an additional measure of out of fit predictive ability and to use pit histograms to check the fit of the Gaussian model to the logged abundance data. 

3. Model selection, variable number and importance for mosquito hazard models

Stepwise selection is used is not generally accepted as being a robust way to select subsets of covariates with full subset selection due to the risks of overfitting or identifying spurious correlations due to remaining collinearity between suites of variables. I appreciate this can be difficult with large number of predictors however and it is reassuring that a relatively small number of environmental predictors remain in the final model. To help the reader judge if the sample size is adequate to test the effects of so many predictors with such a method, the authors should clarify the total number of environmental predictors offered to the model (following exclusion for collinearity) and how this balances out versus the sample size of month/grid square combinations (currently graphed but not specified). Given the limited robustness of stepwise selection and to shed further light on variable importance, I would like to have seen a (supplementary info) table of the changes in WAIC when each variable is dropped from the full “environmental model” until the stopping criteria is reached. A table of environmental variables in the final model, with delta DIC values when each variable is dropped would help to give an indication of variable importance.

4. Model linking predicted JE hazard to outbreak occurrence.

This model seemed very simplistic given the care devoted to the mosquito hazard model. A joint likelihood or hurdle model could have been considered here for outbreak occurrence versus outbreak number. There was no correction for temporal and spatial autocorrelation in model residuals that could lead to over-estimation of vector hazard effects on outbreaks (e.g. in Lines 592-595) 

The accuracy may also be inflated by the large number of absence month-location combinations generated (12,000) to fit the model (though the lack of figures on the number of presence months available makes it difficult to judge the balance in the dataset). The authors should ideally test and present the sensitivity of model results to different ratios of absence to present data. Though the odds ratio for mosquito hazard seem significant (though see comments on autocorrelation), it would be helpful to give some metrics of overall variance explained in outbreak occurrence (e.g. % correct predictions) or sensitivity or specificity to help disease managers to interpret the value of the predictions (does the model do better in some seasons than another for example in this regard).

**Results**

-Does the analysis presented match the analysis plan?

-Are the results clearly and completely presented?

-Are the figures (Tables, Images) of sufficient quality for clarity?

Reviewer #1: See attached review comments

Reviewer #2: See above comments on methods.

The figures are high quality and clear.

**Conclusions**

-Are the conclusions supported by the data presented?

-Are the limitations of analysis clearly described?

-Do the authors discuss how these data can be helpful to advance our understanding of the topic under study?

-Is public health relevance addressed?

Reviewer #1: See attached review comments

Reviewer #2: See above comments on methods for whether the methods and results support the conclusions.

The limitations of the analysis and advances in understanding of the topic are clearly described.

In terms of Public Health relevance, can you say more about practically how the maps could fit into current seasonal management of JE in India to guide which types of intervention intervention, given the levels of uncertainty and accuracy in them and the regional variation in seasonal versus perennial occurrence? 

Lines 604-606. “However, predicted seasonal hotspots in the southeast did not correspond to high cases, potentially revealing the importance of vertical transmission for this disease [28] which requires consistently high vector abundance to maintain the virus in the vector population.” 

This statement is a bit strong, it could also be due to unmeasured environmental factors affecting transmission or the spatial biases in the input data for the model. This should be acknowledged alongside the potential biological explanation. 

Line 626-627. “with the potential for future surveillance efforts to be targeted in those areas with high predicted hazard and a high degree of uncertainty (Fig 3).” I would suggest that you would also need surveillance in high predicted hazard high certainty areas rather than taking these at face value, since you have measured only some types of statistical uncertainty in your analysis and the model needs further validation with independent data.

**Editorial and Data Presentation Modifications?**

Reviewer #1: (No Response)

Reviewer #2: It would be useful to give brief information about disease impacts on communities or the size of JE burdens in the Abstract and Author Summary

It would be useful to outline for context in the introduction where in the landscape or ecosystem human exposure to JE is thought to occur, how is this likely to be influenced by environmental or social factors that would then modulate the relationship between mosquito hazard and spill-over.

Line 109-110 “widely-used SDM approaches, such as boosted regression tree (BRT) models and MaxEnt, do not usually report metrics of predictive uncertainty, or how these vary over space [15,16].”

I think this statement is too strong. The MaxEnt model provides “clamping” metrics indicating areas that are too environmentally disimilar from the training set data to be predicted accurately. Most authors using BRTs, map variation around mean predicted probabilities across runs. 

Line 259: With reference to JE eco-epidemiology in India, can you further explain the decision to clump classes into very broad land use types versus selecting those that are of particularly relevant to mosquito resource use. 

JE Human case data: Clarify briefly what types of location these data from IDSP encompass and how these relate to the location of hazard or exposure in the landscape and the study grain. E.g. these are primary health centre or village locations but will fall into the same 23km square in which exposure occurred. 

Line 316. “ is an occurrence specific intercept”. Clarify what you mean by occurrence specific intercept since the term occurrence could be conflated with spatial location. Is this an observation level random effect for each month by location combination? I may have missed it but I don’t think the priors for this random effect are given later in the methods.

Line 369. Could temporal autocorrelation have been tested for shorter runs of data in simpler ways?

Line 367. WAIC does not measure predictive ability but model parsimony. This sentence should be rephrased, including to inform the reader that lower WAIC = more parsimonious model.

Line 436: “The random selection of inferred absence data points was found to have no substantial (>2) impact on WAIC” This statement is not evidenced anywhere. The WAIC values are not comparable between these models since the underlying dataset is not the same. The best the authors can do is to show (in supplementary) that the delta WAIC values between the baseline, environmental, and seasonal models are still equivalently large. 

Line 461-463: Explain in the methods how you will interpret variable importance of the Bayesian Credible Intervals in the results. 

Line 465-466. “We found that the inclusion of a nonlinear effect for mean monthly temperature without a lag improved model predictive ability when compared to the nonlinear effect with two-month lagged temperature”. This statement is not evidenced by the results presented

Lines 543-544 “assist in guiding interventions when long-term and large spatial scale surveillance data are not available or could not be practically acquired”

Links between rice cultivation and JE Line 586 “lead to changes in local ecology (e.g., biotic interactions such as competition and 587 predation)”

Can you relate these findings to how resource use or dynamics of the key livestock and wildlife hosts for JE might respond to rice cycle dynamics? What types of empirical data would be needed to understand these relationships better?

Line 588 “predicted expansion of flooded areas for rice cultivation [89,90]”. Clarify whether this is a policy driven expansion, or whether you are referring to a climate impact prediction. 

Replace “my” with “we” where it appears.

**Summary and General Comments**

Reviewer #1: See attached review comments

Reviewer #2: Joint likelihood modelling has been widely and successfully applied to model insect taxa to combine abundance and occurrence data or combine data at different scales to understand patterns in biodiversity. This paper represents a novel application to insect vectors and vector-borne disease epidemiology, to maximise insights from sparse occurrence and abundance data and predict hazard from mosquito vectors and risk of outbreaks, focussing on Japanese Encephalitis in India, a key neglected zoonosis there. As such it makes a great contribution to the field and will promote their wider application of these models to vector-borne disease systems and sparse vector surveillance data. The paper is very well written and motivated and should be published once revised. The statistical methods to generate the vector hazard predictions are broadly appropriate, but would benefit from some further clarification and refinement so that the reader can judge the level of support for some of the conclusions drawn. The methods for relating vector hazard to outbreaks seem rather basic and could do with some refinement to support the assertion that hazard is a valuable predictor of outbreaks. I outline these concerns in above along with some minor comments on language and clarify.

PLOS authors have the option to publish the peer review history of their article (what does this mean?). If published, this will include your full peer review and any attached files.

Reviewer #1: No

Reviewer #2: Yes: Beth Purse
---

## [Decision Letter · Decision Letter 1]

24 Jan 2022

Dear Dr Franklinos,

Thank you very much for re-submitting your manuscript "Joint spatiotemporal modelling reveals seasonally dynamic patterns of Japanese encephalitis vector abundance across India" for consideration at PLOS Neglected Tropical Diseases. As with all papers reviewed by the journal, your manuscript was reviewed by members of the editorial board and by several independent reviewers. The reviewers appreciated the attention to an important topic. Based on the reviews, we are likely to accept this manuscript for publication, providing that you modify the manuscript according to the review recommendations.

Sincerely,

Andrew S. Azman

Deputy Editor

Reviewer's Responses to Questions

**Key Review Criteria Required for Acceptance?**

**Methods**

-Are the objectives of the study clearly articulated with a clear testable hypothesis stated?

-Is the study design appropriate to address the stated objectives?

-Is the population clearly described and appropriate for the hypothesis being tested?

-Is the sample size sufficient to ensure adequate power to address the hypothesis being tested?

-Were correct statistical analysis used to support conclusions?

-Are there concerns about ethical or regulatory requirements being met?

Reviewer #1: (No Response)

Reviewer #2: The authors have addressed all of my original comments around the statistical methods. The statistical analysis is highly appropriate and very clearly articulated (e.g. random effects in the model, treatment of vector data, variable selection methods and accuracy/parsimony metrics, sensitivity analysis) and interpreted.

**Results**

-Does the analysis presented match the analysis plan?

-Are the results clearly and completely presented?

-Are the figures (Tables, Images) of sufficient quality for clarity?

Reviewer #1: (No Response)

Reviewer #2: The results and figures are clearly articulated.

**Conclusions**

-Are the conclusions supported by the data presented?

-Are the limitations of analysis clearly described?

-Do the authors discuss how these data can be helpful to advance our understanding of the topic under study?

-Is public health relevance addressed?

Reviewer #1: (No Response)

Reviewer #2: The conclusions are well supported by the data analysis. I appreciated the authors refinement of their statements around the potential policy impact of their work and the additional discussion of the limitations of the analysis.

**Editorial and Data Presentation Modifications?**

Reviewer #1: (No Response)

Reviewer #2: I have some minor comments for the following sentences which the editors/authors can address or not.

Vector abundance i.e., the number of individuals in a site at a given time, and seasonality i.e., intra-annual change in abundance, are important contributors to many epidemiological factors that influence MBD hazard; these factors include pathogen establishment, persistence and transmission [6,8,9].

Please rewrite to clarify the meaning here on how you are categorising the factors affecting MBD hazard. For example, it is strange to have two i.e. in one sentence and the sub-clause “important contributors to many epidemiological factors that influence hazard” is unclear. 

Line 99. Arguably longer vector seasons could also increase contact rates between vectors and hosts, by increasing the duration of seasonal overlap.

Line 309 “exposure to JEV however, other factors”. Suggest splitting into two sentences by inserting full stop after JEV here. 

Line 640 “Surprisingly, we found that the annual area under rice cultivation was negatively associated with vector abundance”. Could clarify that this was not significant.

**Summary and General Comments**

Reviewer #1: Please see attached

Reviewer #2: The authors have made substantial efforts to address all the comments and the paper will make an excellent contribution the the field.

PLOS authors have the option to publish the peer review history of their article (what does this mean?). If published, this will include your full peer review and any attached files.

Reviewer #1: No

Reviewer #2: Yes: Beth Purse

Figure Files:

Data Requirements:

Reproducibility:

References

---

## [Editor Report · Decision Letter 2]

1 Feb 2022

Dear Dr Franklinos,

We are pleased to inform you that your manuscript 'Joint spatiotemporal modelling reveals seasonally dynamic patterns of Japanese encephalitis vector abundance across India' has been provisionally accepted for publication in PLOS Neglected Tropical Diseases.

Best regards,

Andrew S. Azman

Deputy Editor

---

## [Editor Report · Acceptance letter]

16 Feb 2022

Dear Dr Franklinos,

We are delighted to inform you that your manuscript, "Joint spatiotemporal modelling reveals seasonally dynamic patterns of Japanese encephalitis vector abundance across India," has been formally accepted for publication in PLOS Neglected Tropical Diseases.

Best regards,

Shaden Kamhawi

co-Editor-in-Chief

Paul Brindley

co-Editor-in-Chief
